# Antiglycation Effects of Adlay Seed and Its Active Polyphenol Compounds: An In Vitro Study

**DOI:** 10.3390/molecules27196729

**Published:** 2022-10-09

**Authors:** Cheng-Pei Chung, Shih-Min Hsia, Wen-Szu Chang, Din-Wen Huang, Wen-Chang Chiang, Mohamed Ali, Ming-Yi Lee, Chi-Hao Wu

**Affiliations:** 1Department of Nutrition and Health Sciences, College of Human Ecology, Chang Gung University of Science and Technology, Taoyuan 333324, Taiwan; 2Research Center for Food and Cosmetic Safety, College of Human Ecology, Chang Gung University of Science and Technology, Taoyuan 333324, Taiwan; 3School of Nutrition and Health Sciences, College of Nutrition, Taipei Medical University, Taipei 11031, Taiwan; 4Graduate Institute of Metabolism and Obesity, College of Nutrition, Taipei Medical University, Taipei 110301, Taiwan; 5School of Food and Safety, Taipei Medical University, Taipei 110301, Taiwan; 6Nutrition Research Center, Taipei Medical University Hospital, Taipei 110301, Taiwan; 7College of Bioresources and Agriculture, National Taiwan University, Taipei 10617, Taiwan; 8School of Life Science, Huizhou University, No. 46 Yanda Road, Huizhou 516007, China; 9Clinical Pharmacy Department, Faculty of Pharmacy, Ain Shams University, Cairo 11566, Egypt; 10Research Center for Chinese Herbal Medicine, College of Human Ecology, Chang Gung University of Science and Technology, Taoyuan 333324, Taiwan; 11Graduate Programs of Nutrition Science, School of Life Science, National Taiwan Normal University, Taipei 106209, Taiwan

**Keywords:** adlay, adlay testa, antiglycation, phenolic acid

## Abstract

This study aimed to evaluate the antiglycation effects of adlay on protein glycation using in vitro glycation assays. Adlay seed was divided into the following four parts: the hull (AH), testa (AT), bran (AB), and polished adlay (PA). A solvent extraction technique and column chromatography were utilized to investigate the active fractions and components of adlay. Based on a BSA-glucose assay, the ethanolic extracts of AT (ATE) and AB (ABE) revealed a greater capacity to inhibit protein glycation. ATE was further consecutively partitioned into four solvent fractions with *n*-hexane, ethyl acetate (ATE-Ea), 1-butanol (ATE-BuOH), and water. ATE-BuOH and -Ea show marked inhibition of glucose-mediated glycation. Medium–high polarity subfractions eluted from ATE-BuOH below 50% methanol with Diaion HP-20, ATE-BuOH-c to -f, exhibited superior antiglycation activity, with a maximum inhibitory percentage of 88%. Two phenolic compounds, chlorogenic acid and ferulic acid, identified in ATE-BuOH with HPLC, exhibited potent inhibition of the individual stage of protein glycation and its subsequent crosslinking, as evaluated by the BSA-glucose assay, BS-methylglyoxal (MGO) assay, and G.K. peptide-ribose assay. In conclusion, this study demonstrated the antiglycation properties of ATE in vitro that suggest a beneficial effect in targeting hyperglycemia-mediated protein modification.

## 1. Introduction

Advanced glycation end products (AGEs) are a heterogeneous group of reactive, crosslinking compounds produced from nonenzymatic glycation, also known as the Maillard reaction, which occurs between reducing sugars and the amino groups of proteins, nucleic acids, and phospholipids [1]. Protein glycation is formed from Schiff bases, followed by Amadori rearrangements. Amadori products are the initial products of AGEs and can undergo additional glycoxidative modifications [2]. In addition to the above reaction, glucose and Schiff bases can undergo auto-oxidation to form reactive 1,2-dicarbonyl compounds, such as methylglyoxal (MGO) and 3-deoxyglucosone [1,3]. Amadori products also undergo glycoxidation to yield glyoxal and glucosone [2]. The formation of AGEs is relatively slow under physiological status, but significantly accelerated in hyperglycemia conditions. Thus, AGEs may play an important role in the pathogenesis of diabetic complications and aging-related diseases [3,4], and may even have an undefined relationship with COVID-19 morbidity and mortality [5].

AGEs adversely affect our body through several mechanisms [2,3]. The first is the modification of intracellular proteins, including the protein regulation of gene transcription. The second mechanism is through these AGE precursors that diffuse out of the cell and modify the extracellular matrix. These modifications dampen the signaling transduction between the matrix and cells, leading to cellular dysfunction. Finally, the third mechanism is these AGE precursors that modify circulating proteins in the blood, such as albumin. The glycated proteins then bind to the receptor for AGEs (RAGE) and activate AGE–RAGE signaling pathways, evoking oxidative stress and an inflammatory reaction [6].

There are several known mechanisms of AGE inhibition [6]. First, AGE formation can be reduced by tight glycemic control. Carbonyl-trapping agents can block AGE formation, reducing the harmful effects of reactive carbonyl species (RCS). Metal ion chelators can also reduce glycoxidative stress due to suppressing the redox reaction, and finally, AGE levels in vivo can be decreased by crosslink breakers. Aminoguanidine (AG), an AGE inhibitor, is often used as a positive control in antiglycation studies. AG is a nucleophilic agent that traps RCS, such as MGO, by forming non-toxic stable adducts [7]. On the other hand, phenolic antioxidants have demonstrated antiglycative properties primarily by scavenging free radicals and chelating metal ions [6,8]. Theoretically, if an inhibitor possesses more than one of the mechanisms mentioned above, it would make an ideal AGE inhibitor for mitigating diabetic complications and other chronic diseases [9].

Adlay (*Coix lachryma-jobi* L. var. *ma-yuen Stapf*), commonly known as Job’s tears, is widely cultivated in Asia and is utilized as a Chinese folk medicine, as well as a nutritious food [10]. Structurally, adlay seeds consist of the following four parts from the outside to the inside: the hull (AH), testa (AT), bran (AB), and polished adlay (PA). Recent studies have indicated that adlay and its solvent extracts consist of more than 30 ingredients with 20 biofunctional effects, based on clinical and experimental studies [10,11,12,13,14,15,16,17]. In addition, adlay has been shown to regulate blood sugar [14], blood pressure [18], immunity [19], uterine contractions [20], anti-influenza viruses [21], and osteoporosis preventive activities [12]. However, there are still limited studies that explore the effect of adlay on protein glycation and AGE formation. Thus, this study aimed to evaluate the antiglycation potential of adlay using in vitro glycation assays and to investigate the active fractions and compounds of adlay.

## 2. Results

### 2.1. Effects of the Individual Parts of Adlay on Protein Glycation According to BSA-Glucose Assay

In the BSA-glucose assay, glucose was used as the glycating agent, and BSA served as the amine group donor, as the glycated target of glucose. This assay aimed to determine whether adlay could inhibit post-Amadori glycation based on the development of AGE-related fluorescence [8]. Firstly, the adlay seeds were divided into the following four parts: hull (AH), testa (AT), bran (AB), and polished adlay (PA). A solvent extraction technique with column chromatography was used to investigate the active fractions and components of adlay (Figure 1). Ethanol extracts of the hull, testa, bran, and polished adlay were referred to as AHE, ATE, ABE, and PAE, respectively. The results showed that among the individual parts of the adlay, ATE and ABE demonstrated greater inhibitory capacities against protein glycation (Figure 2).

### 2.2. Effects of ATE and ABE Subfractions on Protein Glycation

ATE and ABE were further consecutively partitioned into H_2_O, 1-butanol (BuOH), ethyl acetate (Ea), and *n*-hexane (Hex) fractions. As shown in Figure 3, ATE-BuOH and ATE-Ea exhibited greater antiglycation properties than the other fractions (Figure 3a), and the same findings were found for the ABE fractions (Figure 3b). The ATE-BuOH fractions were obtained and chromatographically isolated to subfractions (a–f), whereas the ATE-Ea fractions were isolated to subfractions (a–h). Inhibitory percentages of ATE-BuOH subfractions -a to -f against protein glycation were as follows: −7%, 0%, 88%, 85%, 56%, and 53%, respectively, at a concentration of 250 μg/mL (Figure 4a). Similarly, the inhibitory potency of the ATE-Ea subfractions (a to h) was 15% to 37% (Figure 4b). In addition, the effects of ATE-BuOH-C and ATE-BuOH-D on the middle stage of glycation were determined by the BSA-MGO assay. MGO belongs to the group of RCS, which is a critical precursor in the formation of AGEs. Figure 5 shows that ATE-BuOH-c and ATE-BuOH-d exhibited significant inhibition of 26% and 30%, respectively, at a concentration of 250 μg/mL. These data suggested the RCS-trapping capacities of ATE-BuOH-c and -d and the main antiglycation components that possibly existed in the ATE-BuOH subfractions.

### 2.3. Effect of ATE-BuOH-Containing Phenolics on the Individual Stage of Protein Glycation

According to our previous study, phenolic acids were major components in ATE-BuOH [22]. In addition, medium–high polarity ethanol extracts of the hull, testa, branty subfractions, eluted from ATE-BuOH below 50% methanol with Diaion HP-20 resin, possessed more significant antiglycation (Figure 4a) as compared with ATE-EA, especially ATE-BuOH-c and –d (Figure 5). 

A high-performance liquid chromatography (HPLC) analysis was carried out to investigate the chemical composition of ATE-BuOH. As shown in Figure 6, chlorogenic acid, caffeic acid, *p*-coumaric acid, and ferulic acid were identified in ATE-BuOH with contents of 1.01 ± 0.03, 1.32 ± 0.04, 9.51 ± 0.94, and 2.54 ± 0.68 mg/g, respectively. However, gallic acid was not detectable in ATE-BuOH.

In the BSA-glucose assay, chlorogenic acid and ferulic acid exhibited 20% and 28% inhibitory activity, respectively (Figure 7), indicating that these phenolics aid in reducing glucose-mediated protein modification. The BSA-MGO assay also determined the inhibition of MGO-mediated protein glycation by these phenolics. The results showed that chlorogenic acid and ferulic acid exhibited significant inhibition of 24% and 15%, respectively. Meanwhile, *p*-coumaric acid had a slight impact on antiglycation (Figure 7). In contrast, the other phenolic compounds identified in ATE-BuOH, such as caffeic acid and 6-methoxy-2-benzoxazolinone [22], showed no antiglycation effect at a concentration of 100 μM (data not shown).

G.K. peptide-ribose assay was used to generate peptides with advanced Maillard reaction products with dimerization through lysine-lysine crosslinking [23]. Rahbar et al. [23] and our previous study [8] demonstrated that incubation of G.K. peptides with ribose resulted in late glycation product formation. Therefore, the present study utilized this model system to evaluate the inhibitory effect of phenolics on protein crosslinking. As shown in Figure 7 (lower panel), chlorogenic acid and ferulic acid exhibited substantial anti-crosslinking activities (47% and 43%, respectively, at a concentration of 100 μM). 

## 3. Discussion

Protein glycation and subsequent AGE formation in the body have been evidenced as a risk factor in the development of diabetic macrovascular and microvascular complications and age-related diseases [24,25]. Clinical observation has revealed that patients with complicated diabetes have 40 to 100% higher AGE levels than healthy subjects [26]. Therefore, investigating AGE inhibitors, especially natural anti-glycation agents with fewer adverse effects, may be a beneficial approach to preventing diabetic complications.

The main question addressed by this study was whether adlay could inhibit protein glycation. Adlay seeds have long been consumed as a food supplement and herbal medicine in traditional Chinese medicine [10,12]. Although the health-promoting effects and therapeutic potential of adlay have been reported [10,12], studies on the potential of adlay to act against protein glycation and AGE formation are limited. This study used a classic in vitro glycation assay to evaluate the effect of individual parts of adlay (hull, testa, bran, and polished adlay) on glucose-mediated BSA glycation. Ethanol was chosen as the initial extraction solvent based on safety considerations for human consumption [27]. Moreover, most active ingredients, including polyphenols, phytosterols, and coixol, found in adlay have been extracted by ethanol with a high yield in previous studies [22,28,29]. For the first time, this study demonstrated that ATE and ABE exhibited better glycation inhibitory effects (Figure 2), suggesting antiglycating agents are possibly present in the bran and testa of adlay. 

Using a solvent extraction technique, two ATE-BuOH subfractions, ATE-BuOH-C and ATE-BuOH-D, exhibited superior antiglycation activities (Figure 4a and Figure 5). Phenolic compounds in adlay seed were analyzed by HPLC in our previous studies [13,22,28,29]. Caffeic acid, chlorogenic acid, *p*-coumaric acid, ferulic acid, gallic acid, *p*-hydroxybenzoic acid, syringic acid, and vanillic acid were identified in ATE-BuOH and/or ATE-Ea fractions. In the present investigation, *p*-coumaric acid (9.51 ± 0.94 mg/g), along with chlorogenic acid (1.01 ± 0.03 mg/g), caffeic acid (1.32 ± 0.04 mg/g), and ferulic acid (2.54 ± 0.68 mg/g) were identified in ATE-BuOH through HPLC analysis (Figure 6). Notably, chlorogenic acid and ferulic acid from ATE-BuOH exhibited significant potent inhibition of the individual stage of protein glycation, especially in the reduction in protein crosslinking (Figure 7). After partially purified by column chromatography, the ATE-BuOH-c and –d (eluent from 50 to 25% methanol) showed superior inhibitory effects on antiglycation (Figure 3a, Figure 4a and Figure 5). A previous study has demonstrated that the major components in subfractions eluted from 50 to 25% methanol from ATE-BuOH were caffeic acid (9.02 mg/g subfractions), chlorogenic acid (30.30 mg/g subfractions), and ferulic acid (0.05 mg/g subfractions) [22]. These results suggest that the antiglycation properties of ATE were at least partly related to its phenolic acid content.

Research has demonstrated that antioxidative polyphenols show potent antiglycation activities [1,6,8]. Several studies have drawn attention to the positive correlation between the free radical scavenging activity and antiglycation capacity, which may be due to the interruption of ROS formation during glycation [6,8]. In addition, the antioxidant ability of phenolic acids, such as caffeic acid and chlorogenic acid, depends on the number of hydroxyl groups in the molecule that would be strengthened by steric hindrance [30]. The present study showed that chlorogenic acid and ferulic acid are potent inhibitors that act at the individual stage of glycation, including post-Amadori glycation, as evidenced by the decreased development of AGE-related fluorescence in BSA-glucose assay (Figure 7, upper panel). Notably, these phenolic acids exhibited MGO-trapping activity and anti-crosslinking action, as evidenced by the MGO-BSA assay (Figure 7, middle panel) and G.K. peptide-ribose assay (Figure 7, bottom panel). However, certain compounds, such as caffeic acid, had no inhibitory effect on glycation (data not shown), which was consistent with a former study [31]. 

Since MGO is a critical intermediate and precursor during AGE formation, the MGO-BSA assay was utilized to determine the middle stage of protein glycation. MGO served as the glycating agent, and BSA provided the amine source targeted by the glycating agents. Albumin was used as the protein because it is the most abundant protein in serum. A clinical study noted that the serum levels of glycated albumins are two to three times higher in diabetic patients than in healthy people [32]. It is known that MGO can readily react with proteins that contain lysine residues and crosslink with proteins, along with ROS production [33]. MGO-glycated BSA was detected by specific fluorescence formation. This study showed that ATE-BuOH subfractions and their active phenolics (chlorogenic acid and ferulic acid) achieved nearly 30% inhibition of MGO-mediated protein modification (Figure 5 and Figure 7), indicating the possible MGO-trapping potential of adlay.

Nagaraj and his colleagues [34] indicated that protein crosslinking is the major end result of the Maillard reaction. In this study, a synthetic G.K. peptide that contained lysine residue was incubated with ribose. This chemical model was designed to generate peptides with AGEs that dimerize through lysine-lysine crosslinking and increase the late glycation product formation, as determined by intrinsic fluorescence formation [23]. Chlorogenic acid and ferulic acid were found to be anti-crosslinking agents (Figure 7, lower panel). Taken together, the data suggested that the presence of methoxy groups at the C3 position of the aromatic ring in ferulic acid and the quinic acid groups in chlorogenic acid may contribute to the antiglycation actions.

## 4. Materials and Methods

### 4.1. Preparation of Adlay Extracts

The adlay utilized in this experiment was purchased from a local farmer in Taichung, and the variety of adlay was Taichung Shuenyu no. 4 (TCS 4) of *Coix lachryma-jobi* L. var. *ma-yuen Stapf*. Sample preparation and extraction used previously reported methods from our laboratory [22]. Briefly, adlay seeds were air-dried at 40 °C and ground to obtain AH, and further ground to separate dehulled adlay and AT by an electric fan. After these procedures, the whole grain contained 7.2 ± 0.4% water, while that in AH and AT contained 5.2 ± 0.1% and 7.2 ± 0.1%, respectively. A total of 12 kg of AT was extracted with 120 L of ethanol 3 times at room temperature for 24 h, and filtered through #1 filter paper (Cytiva, Washington, D.C., USA). The filtrate was then concentrated to dryness to yield approximately 538 g of dried ATE (Figure 1). The dried extracts were stored at −20 °C until use. ATE was then suspended in a 10% methanol solution and partitioned with *n*-hexane to obtain the lipid-rich ATE-Hex fraction, yielding approximately 144 g. The defatted ATE was further partitioned with Ea to obtain about 160 g of ATE-Ea. The remaining ATE was partitioned with butanol to obtain about 50 g of ATE-BuOH. Finally, the residual extract was the ATE-H_2_O fraction. ATE-Ea was dissolved in Ea and subjected to column chromatography on a silica gel to yield ATE-Ea subfractions a–h. The gradient used for elution was a hexane/Ea gradient from 0 to 100% EA (every 10%), and the sub-fractions with similar chemical compositions on thin-layer chromatography (TLC) were combined and concentrated to dryness under a vacuum at 60 °C. Subfractions were stored at −20 °C until use. On the other hand, ATE-BuOH was dissolved in methanol and subjected to column chromatography on a Diaion HP-20 resin to yield ATE-BuOH subfractions a–f. The gradient used for elution was an H_2_O/MeOH gradient from 0 to 100% MeOH (every 25%) and was washed with 100% EA. Sub-fractions with similar chemical compositions on TLC were combined and concentrated to dryness in a vacuum at 60 °C and stored at −20 °C until use.

### 4.2. BSA-Glucose Assay

The BSA-glucose assay was devised by Matsuura et al. [35] to screen for effective AGE inhibitors from natural product extracts. A total of 50 mL of 50 mM phosphate buffer (pH 7.4) and 200 mM glucose were mixed as a stock solution. Then, 500 μL of reaction mixture containing 400 μg of BSA and 200 mM glucose, with or without 10 μL of plant extracts, in 50 mM phosphate buffer at a pH of 7.4 in an Eppendorf tube was prepared. The reaction mixture was heated at 60 °C on a heat block for 48 h. A blank, unreacted solution sample without inhibitors was kept at 4 °C until measurement. Samples were cooled to room temperature. After cooling, 100 μL aliquots were transferred to new 1.5 mL plastic tubes, and 10 μL of TCA was added to each tube. The tubes were centrifuged at 15,000 rpm at 4 °C for 4 min. The supernatant was removed and AGE-BSA precipitate was dissolved in 800 μL of PBS. The fluorescence of the samples was measured at an excitation wavelength of 370 nm and an emission wavelength of 440 nm, and inhibitory activity was calculated as a percentage using the following formula:I% = [1 − (fluorescence of BSA + glucose + inhibitor − fluorescence of BSA + inhibitor)/(fluorescence of BSA + glucose) − (fluorescence of BSA)] × 100%

### 4.3. BSA-MGO Assay

The middle stage of protein glycation was determined using the BSA-MGO assay described by Wu et al. [8], with a slight modification. BSA (50 mg/mL) was incubated with 100 mM MGO under sterile conditions in 0.1 M phosphate buffer (pH 7.4) at 37 °C for 9 days. Sample fractions and subfractions were added to the model system at final concentrations of 125 μg/mL and 250 μg/mL, respectively. The sample solutions were kept at 4 °C until measurement. After cooling, 100 μL of aliquots were transferred to 1.5 mL plastic tubes, and 10 μL of 100% TCA was added to each tube. Tubes were centrifuged at 15,000 rpm at 4 °C for 4 min. The supernatant was removed, and the precipitate of MGO-BSA was dissolved in 800 μL of PBS. The fluorescence of the samples was measured at the excitation and emission maxima of 330 and 410 nm, respectively, and compared with that of the unincubated blank that contained the protein, MGO, and inhibitors. The percent inhibition was calculated as follows:I% = [1 − (fluorescence of BSA + MGO + inhibitor − fluorescence of BSA + inhibitor)/(fluorescence of BSA + MGO) − (fluorescence of BSA)] × 100%(1)

### 4.4. G.K. Peptide−Ribose Assay

This test was used to evaluate the ability of flavonoids to inhibit the crosslinking of G.K. peptides (last glycation products) in the presence of ribose, using the method described by Rahbar et al. [23]. The G.K. peptide (80 mg/mL) was incubated with 0.8 M ribose under sterile conditions in 0.5 M sodium phosphate buffer (pH 7.4) at 37 °C for 24 h. The adlay samples or phenolic acids were added at final concentrations of 125 μg/mL and 250 μg/mL, respectively. At the end of the incubation period, samples were analyzed for specific fluorescence (excitation, 340 nm; emission, 420 nm). The % inhibition by different concentrations of inhibitor was calculated as described above.

### 4.5. HPLC Analysis

The HPLC analysis method according to a previous report [22], with a slight modification, was used and carried out on an UltiMate 3000 pump equipped with an autosampler column compartment system and a variable wavelength detector (DIONEX, manufacturer, Sunnyvale, CA, USA). A 4.6 mmØ × 250 mm (5 µM) reverse-phase C18 column was used (HiQ sil C18HS). Gradient elution was performed with 2% acetic acid_(aq)_ (*v*/*v*, solvent A) and 0.5% acetic acid_(aq)_/acetonitrile (1:1, *v*/*v*, solvent B) at a constant flow rate of 1 mL/min. The initial condition was at 5% B, and the eluent was as follow: 0–10 min B increased from 5 to 10%, 10–40 min B increased from 10 to 40% and 40–55 min B increased from 40 to 55%. The waste was washed by increasing B from 55 to 100% within 10 min, and the eluent was adjusted to the initial condition (5% B) within 15 min. The absorbance at 280 nm was recorded. Phenolic standards, dissolved in DMSO, were analyzed, and peak areas of serial diluted standards were calculated for calibration curves. ATE-BuOH was analyzed using the HPLC system, and the contents of phenolic acids were compared with the retention times and calibration curves.

### 4.6. Statistics

All the experiments were performed in triplicate, and the values are expressed as the mean ± SD. Statistical analysis was performed via one-way analysis of variance (ANOVA), and multiple comparisons were performed utilizing Duncan’s multiple range test. All statistical analyses were performed using the computer software SPSS 12.0. *p* < 0.05 was considered statistically significant.

## 5. Conclusions

Adlay testa exhibits antiglycative activities that affect protein glycation and the protein’s subsequent crosslinking. Adlay inhibits glycation, perhaps mainly due to its antioxidant phenolic acid compounds of chlorogenic acid and ferulic acid. Therefore, adlay seeds may be a potential candidate for the future development of alternative therapeutics for AGE-related diseases.

## Figures and Tables

**Figure 1 molecules-27-06729-f001:**
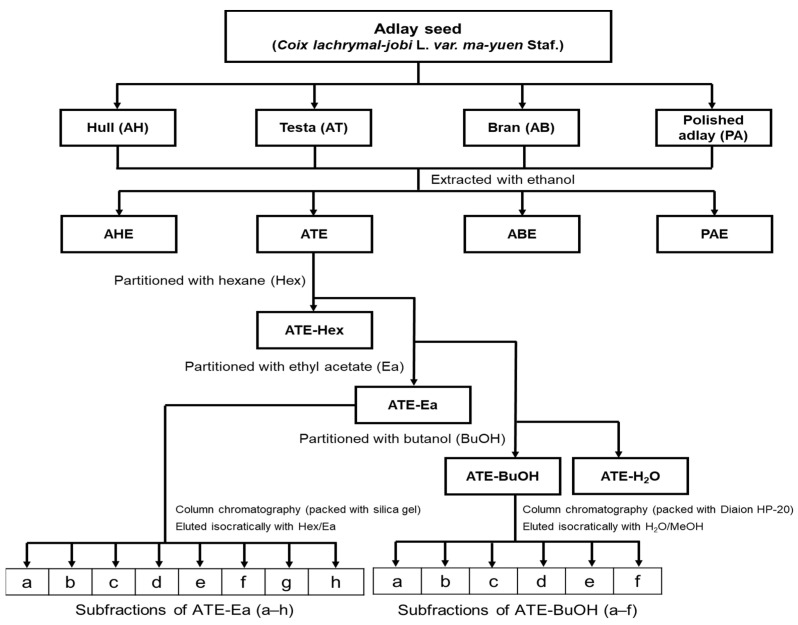
Partitioning scheme for preparing and isolating active fractions and components from adlay ethanolic extracts. ATE-Ea was dissolved in Ea and subjected to column chromatography on a silica gel with a Hex/Ea gradient from 0 to 100% EA (every 10%) to yield a–h. ATE-BuOH was dissolved in MeOH and subjected to column chromatography on a Diaion HP-20 resin with an H_2_O/MeOH gradient from 0 to 100% MeOH (every 25%)to yield a–f.

**Figure 2 molecules-27-06729-f002:**
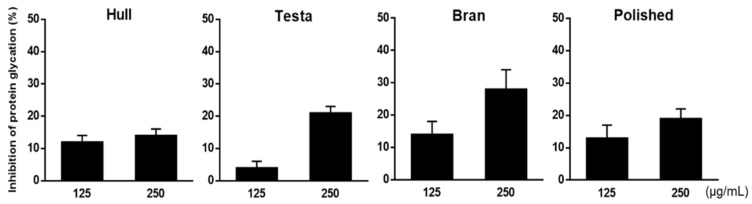
Effects of ethanolic extracts of adlay hull (AHE), adlay testa (ATE), adlay bran (ABE), and polished adlay (PAE) on glucose-mediated development of fluorescence of AGEs (BSA-glucose assay). Results are means ± SD for *n* = 3.

**Figure 3 molecules-27-06729-f003:**
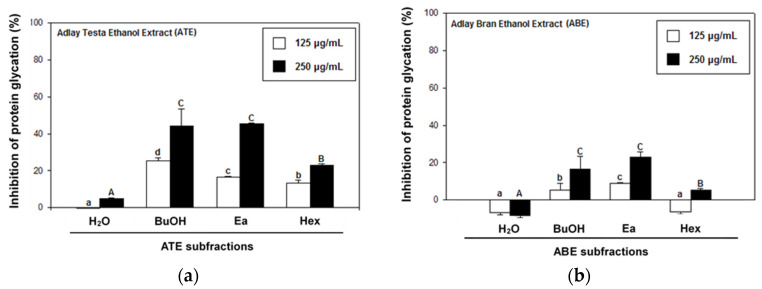
Effects of different solvent fractions of ATE (**a**) and ABE (**b**) on glucose-mediated development of fluorescence of AGEs (BSA-glucose assay). Groups with different letter superscripts are significantly different (*p* < 0.05). a–c, treated with 125 μg/mL; A–C, treated with 250 μg/mL.

**Figure 4 molecules-27-06729-f004:**
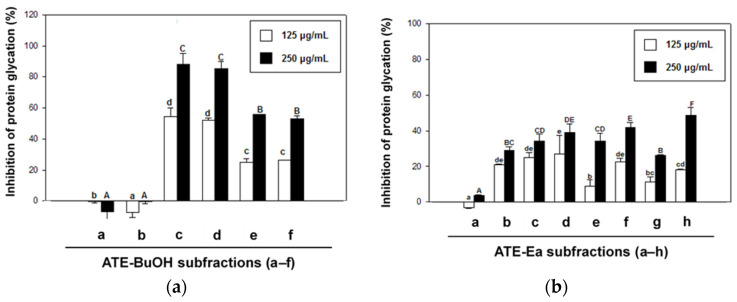
Effects of ATE-BuOH (**a**) and ATE-Ea (**b**) subfractions on glucose-mediated development of fluorescence of AGEs (BSA-glucose assay). Results are means ± SD for *n* = 3. Groups with different letter superscripts are significantly different (*p* < 0.05). a–h, treated with 125 μg/mL; A–F, treated with 250 μg/mL.

**Figure 5 molecules-27-06729-f005:**
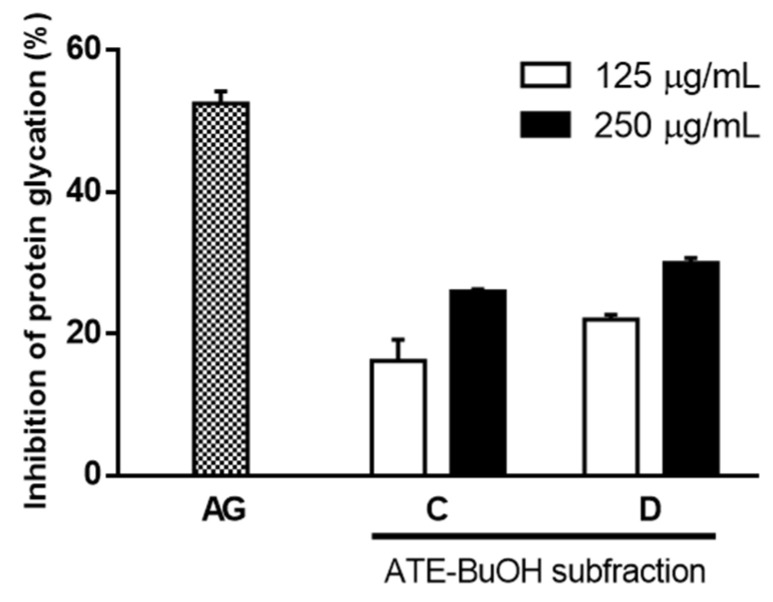
Effects of ATE-BuOH subfractions C and D on MGO-mediated development of fluorescence of AGEs (BSA-MGO assay). Results are means ± SD for *n* = 3. AG, aminoguanidine. C, the subfraction C of ATE-BuOH. D, the subfraction D of ATE-BuOH.

**Figure 6 molecules-27-06729-f006:**
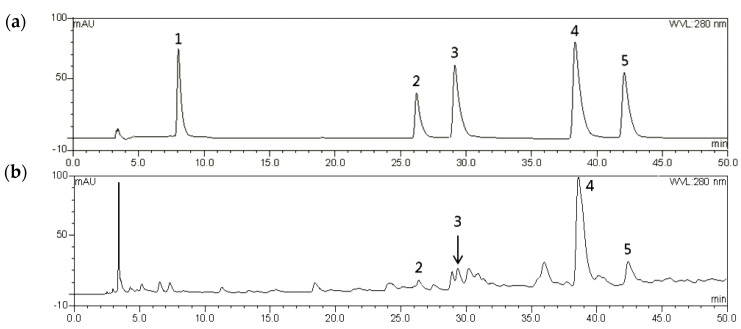
High−performance liquid chromatography (HPLC) chromatographic profiles recorded at 280 nm of (**a**) phenolic compound mixed standards (each 0.1 mg/mL) and (**b**) the ATE-BuOH fraction (10 m(g/mL) (1: gallic acid, 2: chlorogenic acid, 3: caffeic acid, 4: p-coumaric acid, and 5: ferulic acid).

**Figure 7 molecules-27-06729-f007:**
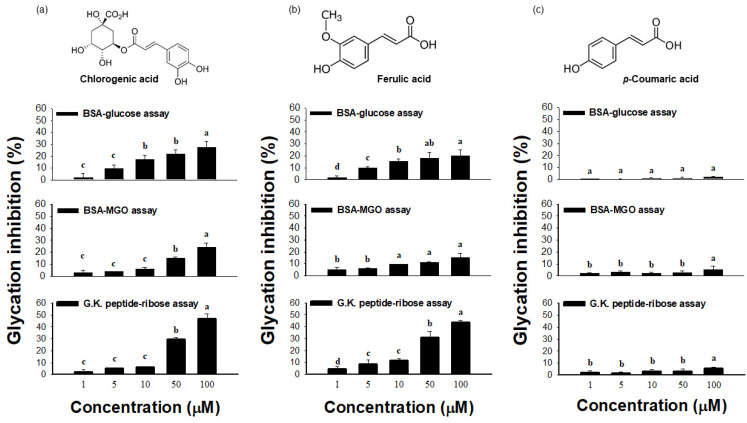
Chemical structures and the effects of chlorogenic acid (**a**), ferulic acid (**b**), and *p*-coumaric acid (**c**) identified in ATE-BuOH fractions on protein glycation and crosslinking. Dose responses of glycation inhibition (%) on individual stages of protein glycation were determined by model systems of the BSA-glucose assay (upper panel), the BSA-MGO assay (middle panel), and the G.K.-ribose assay (bottom panel). Data are the means ± SD for *n* = 6. Groups with different letters (a–d) are significantly different from each other in individual assays (*p* < 0.05).

## Data Availability

Data are contained within the article.

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
