# Peer review of "Antiglycation Effects of Adlay Seed and Its Active Polyphenol Compounds: An In Vitro Study"

_molecules, 2022, doi:10.3390/molecules27196729_

Round 1
Reviewer 1 Report
This is an in vitro study which deals with the antiglycation effects of adlay seed and its active polyphenol compounds. The study is interesting, easy to understand since it has a flow, and its purpose is clearly stated. However, there are some points that need clarification:
· -What was the water content of the seeds?
· -Pg2, ln 94: I believe the PAE is APE.
· -Pg3, ln 104: The authors make no comments about the negative percentage. How do they explain this result?
· - Figure 1: The figure needs correction since the authors extracted the four adley seeds parts and not the whole seeds. Thus, an extra step before the extracts (AHE, ATE etc) is needed.
· -The figure legends must be shortened. The methods are already described in the M&M part.
· -The different percentages of gradients used should be stated (pg 7&8).
· -Pg 8, ln 274: The number “ten” should be written as “10” for consistency reasons.
· -A revision in English in needed.
Reviewer 2 Report
Comments:
In this manuscript, the authors described “Antiglycation effects of adlay seed and its active polyphenol compounds: in vitro study”. This paper show that this study demonstrates the antiglycation properties of ethanol extracts of adlay testa (ATE) in vitro that suggest a beneficial effect in targeting hyperglycemia mediated protein modification. However, there are a few points that need to be clarified.
Comment
1. The author shall be show the HPLC fingerprint of the extract of Adlay seed.
2. BSA-glucose assay, BSA-MGO assay and G.K.-ribose assay. The authors need to compare the differences of these three methods in detail.
3. The ATE-Ea subfraction also works well for BSA-glucose assay. What are its active ingredients of antiglycation effects?
Round 2
Reviewer 2 Report
accepted
Author Response
Thanks for reviewer comments.